# Vaccination and Factors Related to the Clinical Outcome of COVID-19 in Healthcare Workers—A Romanian Front-Line Hospital’s Experience

**DOI:** 10.3390/vaccines11050899

**Published:** 2023-04-25

**Authors:** Carmen-Daniela Chivu, Maria-Dorina Crăciun, Daniela Pițigoi, Victoria Aramă, Monica Luminița Luminos, Gheorghiță Jugulete, Cătălin Gabriel Apostolescu, Adrian Streinu Cercel

**Affiliations:** 1Department of Epidemiology 1, Carol Davila University of Medicine and Pharmacy, 050474 Bucharest, Romania; carmen-daniela.chivu@drd.umfcd.ro (C.-D.C.); daniela.pitigoi@umfcd.ro (D.P.); 2Emergency Clinical Hospital for Children “Grigore Alexandrescu”, 011743 Bucharest, Romania; 3National Institute for Infectious Diseases “Prof. Dr. Matei Balș”, 021105 Bucharest, Romania; victoria.arama@umfcd.ro (V.A.); luminita.luminos@umfcd.ro (M.L.L.); gheorghita.jugulete@umfcd.ro (G.J.); catalin-gabriel.apostolescu@drd.umfcd.ro (C.G.A.); adrian.streinucercel@umfcd.ro (A.S.C.); 4Department of Infectious Diseases 1, Carol Davila University of Medicine and Pharmacy, 020021 Bucharest, Romania; 5Department of Infectious Diseases 3, Carol Davila University of Medicine and Pharmacy, 020021 Bucharest, Romania

**Keywords:** COVID-19, healthcare workers, clinical outcomes, exposure, individual risk, vaccination

## Abstract

The study aims to describe the frequency of COVID-19 in healthcare workers (HCWs) in a designated hospital for COVID-19 treatment in Bucharest, Romania, and to explore COVID-19 vaccination and other factors associated with the clinical outcome. We actively surveyed all HCWs from 26 February 2020 to 31 December 2021. Cases were laboratory-confirmed with RT-PCR or rapid test antigen. Epidemiological, demographic, clinical outcomes, vaccination status, and co-morbidities data were collected. Data were analyzed using Microsoft Excel, SPSS, and MedCalc. A total of 490 cases of COVID-19 in HCWs were diagnosed. The comparison groups were related to the severity of the clinical outcome: the non-severe group (279, 64.65%) included mild and asymptomatic cases, and the potentially severe group included moderate and severe cases. Significant differences between groups were registered for high-risk departments (*p* = 0.0003), exposure to COVID-19 patients (*p* = 0.0003, vaccination (*p* = 0.0003), and the presence of co-morbidities (*p* < 0.0001). Age, obesity, anemia, and exposure to COVID-19 patients predicted the severity of the clinical outcomes (χ^2^ (4, *n* = 425) = 65.69, *p* < 0.001). The strongest predictors were anemia and obesity (OR 5.82 and 4.94, respectively). In HCWs, mild COVID-19 cases were more frequent than severe cases. Vaccination history, exposure, and individual risk influenced the clinical outcome suggesting that measures to protect HCWs and occupational medicine are important for pandemic preparedness.

## 1. Introduction

The 2019 coronavirus disease (COVID-19) outbreak was first reported in Wuhan, China, and spread rapidly worldwide. As of 1 March 2023, over 750 million confirmed cases and over 6.8 million deaths have been reported globally [1]. Healthcare workers (HCWs) have been at high risk for developing COVID-19 through professional exposure. Reporting HCW status in data on COVID-19 is limited worldwide to 63 countries. Based on these reports, the World Health Organization (WHO) estimated an upper range of deaths in this category of over 180,000 until May 2021 [2].

In Romania, the first COVID-19 case was detected on 26 February 2020. By the end of 2021, weekly reports from the National Center for Surveillance and Control of Communicable Diseases showed a total of 1,707,069 COVID-19 cases in the general population, from which 17,584 (1%) COVID-19 cases were in HCWs [3] Weekly reported COVID-19 cases among the HCWs overlap only partially on the epidemic curve in the general population [3]. The summarized reported data from Romania is presented in the Appendix A.

The WHO, European Centre for Disease Prevention and Control (ECDC), and national health authorities have developed safety protocols for medical personnel, such as appropriate personal protective equipment (PPE) and training for proper use. When the COVID-19 vaccines became available, HCWs were the first category of the population prioritized for vaccination [4,5,6]. According to ECDC, in the European Union (EU) and European Economic Area (EEA), the vaccination coverage with the primary course was 73% for the general population and 90.4% for HCWs [7]. The Romanian COVID-19 vaccination campaign started on 27 December 2020. By the end of 2021, vaccination coverage with the primary course was 42.1% in the general population and 94% in HCWs [8].

The factors associated with the exposure and the correlation between these factors and clinical outcomes in HCWs diagnosed with COVID-19 are still a subject of interest. Previous studies demonstrated the impact of vaccination, exposure, and comorbidities on the clinical severity of COVID-19 in specified populations and the general population, such as cardiovascular disease, obesity, anemia, diabetes, malignancies, and neurological diseases [9,10,11,12,13,14]. 

This study investigated the first episode of SARS-CoV 2 infection in HCWs from a designated hospital for COVID-19 treatment in Bucharest, Romania, to describe the frequency of clinical outcomes of COVID-19 in healthcare workers (HCWs) and to evaluate the factors related to moderate-to-severe COVID-19 outcomes.

## 2. Materials and Methods

### 2.1. Hospital Setting and Participants

We conducted a retrospective cohort study at the National Institute for Infectious Diseases “Professor Dr. Matei Balș” (INBI), the largest infectious diseases hospital from Bucharest, Romania, designated as a reference center for the management of public health alerts in Romania [15,16]. The first COVID-19 case in the country was managed in the hospital setting. Since 7 April 2020, the hospital has admitted only COVID-19 patients, being a frontline hospital for COVID-19 confirmed cases [15].

The hospital has emergency departments, where HCWs evaluate suspected and confirmed cases, intensive care units, and wards for confirmed cases; all these departments were defined as high-risk departments. All other departments were considered low risk. 

The population studied represented HCWs with an employment contract within INBI “Prof. Dr. Matei Balș” and current activity during the studied period as the primary criterion for inclusion. The cases with incomplete clinical data were excluded from the risk analysis.

The HCWs are an essential tool in managing the pandemic. Good management of human resources during the pandemic is crucial, as COVID-19 cases in HCWs were isolated for a variable time interval according to clinical severity. The hospital’s policy was to protect human resources by active surveillance, on-site clinical evaluation and treatment, and applying non-pharmaceutical measures (PPE) and vaccination.

### 2.2. Definition and Data Collection

COVID-19 cases were laboratory confirmed with reverse transcriptase polymerase chain reaction (RT-PCR) or rapid test antigen (RTA) for SARS-CoV 2. A testing protocol was applied to identify the asymptomatic cases. All persons coming from medical leave or holidays and all the exposed contacts to confirmed cases as part of outbreak investigations were tested.

Symptomatic HCWs were tested according to the national surveillance methodology, based on the case definition [17]. The laboratory tests were performed in the healthcare facility or the public health department laboratory network if the debut of symptoms was at home. The national surveillance methodology for COVID-19 was modified regarding the case definition on 4 January 2021, and the confirmed case of COVID-19 was considered a case with RT-PCR or RTA [18]. Before the mentioned date, all cases were confirmed by RT-PCR.

Exposure, epidemiological, demographic information, and vaccination status were prospectively collected for all laboratory-confirmed cases using a standardized WHO questionnaire and an adapted outbreak investigation form. During the study period, the HCWs were diagnosed with COVID-19 once, twice, or three times; the study targeted only the first episode of infection. The history of exposure and the use of personal protective equipment during medical activity was registered by the WHO questionnaire “Risk assessment and management of exposure of health care workers in the context of COVID-19: Interim guidance WHO” taken by one-time telephonic interview, and the information was collected on printed forms [19]. The first part of the questionnaire (Q4A–4D) evaluates exposure with questions about clinical care, face-to-face contact with patients, aerosol procedures, and contact with the environment of patients. The second part (Q 5–7) of the questionnaire evaluates adherence to non-pharmacological protective measures: proper use of personal protective equipment during current medical activities or aerosol-generating procedures and accidents with biological material. Information about the frequency of hand hygiene after contacting confirmed or suspected COVID-19 patients and the frequency of using different types of personal protective equipment (never, sometimes, most of the time, and all the time) was requested from each HCW confirmed with COVID-19. 

The questionnaire was applied by junior physicians who were trainees in field epidemiology, and the time for answering the questions was estimated as 5 min. Medical staff from the Infection Control Department applied the outbreak-adapted investigation form to clarify the exposure and validated the questionnaire. A report to the local Public Health Department was also filled in for each case.

The Microsoft Excel program was used to collect data, and recordings were made retrospectively. 

Medical records were reviewed retrospectively for clinical characteristics and outcomes from the hospital’s electronic system. According to the hospital’s policy, every COVID-19 case in a HCW had at least a one-day admission for clinical and imagistic evaluation. 

The severity of COVID-19 was recorded as the doctor filled in the discharge forms. The clinical outcomes registered were according to the WHO progression scale: mild or asymptomatic, moderate, severe, critical, and death [20]. Only one HCW died on the first day of admission and was excluded from the statistics. 

Data were collected from the hospital’s informatic system to assess the individual risk of developing severe COVID-19. Co-morbidities, such as hypertension, cardiovascular disease, neurological, renal, and oncologic, were registered for all cases. Readmissions were identified, and the clinical outcome was updated accordingly. Data has been validated and matched over multiple sets by a data engineer’s automation in Python.

Vaccinated HCWs were defined as persons who received a minimum of 2 doses of the COVID-19 vaccine. The vaccination status of cases was obtained from the hospital’s infection control department responsible for the COVID-19 vaccination campaign of the hospital’s HCWs and updated from the National Electronic Registry of Vaccinations. 

Characteristics, such as age, gender, vaccination, job category, department of activity, exposure during work activity, the proper use of personal protective equipment, frequency of hand hygiene practice, accidents through exposure to blood and biological products, and co-morbidities, were included in the analysis to evaluate the individual and the occupational risk.

### 2.3. Statistical Analysis

After data processing, two groups were distinguished for comparison: non-severe COVID-19 outcomes, where there were included asymptomatic and mild conditions of COVID-19, and potential moderate-to-severe outcomes, where we included moderate, severe, and critical cases. 

Continuous and categorical variables were presented in mean, median, interquartile ranges, numbers, and percentages. Differences between proportions of categorical variables were assessed using χ^2^ tests in MedCalc.

The logistic regression model was used to calculate the OR and 95% CI for moderate-to-severe infection. A *p*-value < 0.05 was considered significant. When selecting input variables, we restricted the number to avoid overfitting the model and chose only the independent variable [21].

The statistical analysis and data visualization were performed using Microsoft Excel, IBM SPSS for Macintosh, version 29.0.0.0, (Armrok, NY), and MedCalc Statistical Software version 19.2.6 (MedCalc Software bv, Ostend, Belgium; https://www.medcalc.org; accessed 19 April 2023)

### 2.4. Ethics Approval

The study protocol was approved by the Institutional Bioethics Committee with the registration number C02648/16.03.2022.

## 3. Results

From 26 February 2020 to 31 December 2021, 1314 employees worked at INBI and were actively surveyed by the Department of Infection Control. Among them, there were 490 recorded first COVID-19 episodes (37.2%). After validation, 425 cases (87.6%) were included in the analysis to evaluate the exposure and adherence to non-pharmacological measures. Sixty-four cases were excluded from the database due to missing or incomplete data, and one died on the first day of admission.

The flow chart of data exclusion is represented in Figure 1.

The 490 laboratory confirmed cases consisted mainly of women (85.1%), corresponding to the gender structure of the hospital staff. The median age of the cases was 45 (IQR 37–51). 

The age distribution in the entire surveyed population showed that the age group of 40–49 (427; 32.5%) is larger then other groups. When comparing the laboratory-confirmed cases group with the uninfected HCWs group, we found statistically significant differences in the age group of 18–29 (10.4% cases vs. 22.0% uninfected, *p* ≤ 0.0001 χ^2^(1) = 28.4) and in the age group of 40–49 (*p* = 0.0088, χ^2^(1) = 6.85). Professional category distribution showed that the group of nurses was larger (467; 35.5%), and statistical differences between the laboratory-confirmed cases group and uninfected HCWs group were identified in physicians (19.6% cases vs. 27.1%, uninfected, *p* = 0.0023, χ^2^(1) = 9.3). Detailed data are represented in Table 1.

The monthly distribution of COVID-19 cases in HCWs’ first episode is presented in Figure 2.

The comparison groups were created according to the WHO clinical progression scale [20]. The clinical outcomes of COVID-19 in HCWs ranged from asymptomatic to severe, with most of the cases being asymptomatic or mild (279, 64.7%). The moderate and severe cases were included in a group of potentially severe cases, representing a smaller group (146, 34.3%). The differences between non-severe and potentially severe cases are presented in Table 2.

The median age of the 425 HCWs was 45 years (IQR 37–51 years), and 362 (85.2%) were women, corresponding to the gender structure of hospital staff. When comparing the mean, significant differences were seen in age between non-severe and severe cases of HCWs (*p* = 0.0035, SE = 0.95). There were no statistically significant differences between groups concerning job categories. Still, there were identified statistically significant differences between groups regarding exposure, vaccination, and co-morbidities, as shown in Table 2.

Evaluating the adherence to non-pharmacological protective measures (Q 5–7) of HCWs confirmed with COVID-19, we found only 5 (3.4%) HCWs with a high-risk evaluation in the moderate-to-severe group and 24 (8.6%) in the non-severe group.

### 3.1. Vaccination Status

We found that 85 HCWs (20.0%) were vaccinated before the first episode of COVID-19, with significant differences between groups (*p* = 0.0003, χ^2^(1) = 13.126) regarding severity outcome.

### 3.2. Co-Morbidities

Co-morbidities were identified in 107 HCWs (25.2%), more present in the moderate-to-severe group (*p* < 0.0001, χ^2^(1) = 61.082). 

### 3.3. Exposure

The WHO questionnaire evaluated the exposure and adherence to non-pharmacological measures. The exposure to confirmed COVID-19 patients by direct care was evaluated by face-to-face contact within a distance of less than 1 m, the presence in the room when any aerosol-generating procedures were performed on the patient, and direct contact with the patient’s environment. After scoring the first four questions, the HCWs were classified as exposed (63.1%) and non-exposed (36.9%) to COVID-19 during work in the hospital.

Statistical analysis for exposure revealed no differences in the severity of clinical outcomes when aerosol-generating procedures were performed (Table 3).

SARS-CoV-2 sample sequencing was available only for a few cases, and we could not use the data for comparisons. We did compare the severity of the cases to the circulant variant in the country, as reported by the National Public Health Institute. Most of the cases were registered during the circulation of the Wuhan-Hu-1 variant. The results are represented in Figure 3.

### 3.4. Logistic Regression

Logistic regression was performed to assess the probability of developing moderate-to-severe clinical outcomes for COVID-19 in HCW for several factors (Table 4). The model contained four independent variables (age, obesity, anemia, and exposure to COVID-19 patients) that made a unique statistically significant contribution to the model. 

The full model containing all predictors was statistically significant, χ^2^ (4, *n* = 425) = 65.69, *p* < 0.001, indicating that the model was able to distinguish between cases with moderate-to-severe outcomes and cases with mild clinical outcomes. The model as a whole explained between 14.3% (Cox and Snell R square) and 19.3% (Nagelkerke R squared) of the variance of moderate-to-severe outcomes and correctly classified 73.4% of cases. 

The strongest predictors for moderate-to-severe clinical outcomes were anemia and obesity, recording an odds ratio of 5.82 and 4.94, respectively, both with considerably large confidence intervals. These intervals indicate that we must identify other risk factors in these subgroups and complete the research with discrete data on these variables. The odds ratio of 1.04 for age shows that the likelihood of developing a moderate to severe condition is multiplied 1.04 times for every additional year of age. Although vaccination was statistically significant when comparing the non-severe and moderate-to-severe groups, it did not meet the criteria for predictors in the logistic regression.

## 4. Discussion

Active surveillance based on an adapted testing protocol is an excellent method to collect and analyze data about COVID-19 in HCWs. The hospital surveillance system for COVID-19 in HCWs was aimed at early detection, isolation, and epidemiological investigations case by case. These activities were conducted by the Department of Infection Control of the Hospital and reported to Regional Department for Public Health. The testing protocol used in this study included testing HCWs with symptoms [17] and testing asymptomatic individuals, as the relatively high incidences of asymptomatic cases in the general population put an alert for the presence of asymptomatic HCWs [22]. COVID-19 management was based on European and international recommendations, as frontline infectious disease hospital meetings with international experts took place to better understand and apply these recommendations. 

By comparing the differences in epidemiological, demographic, co-morbidities, and exposure data during medical activity, we wanted to give new insights into evaluating the risk of developing non-severe or moderate-to-severe COVID-19 outcomes. This study and other published studies differ in design, population range, and study period. Most studies were published in 2020 and had a shorter study period. During almost two years of active surveillance of HCWs, this study surprised a unitary approach regarding PPE in HCWs and using masks in public spaces for the general population. Variations were the pathogenicity of the virus as we had three evolving variants of SARS-CoV-2 during the study period and the protection offered by vaccination and other biological products.

In this study, we actively surveyed all hospital employees and investigated 425 cases of COVID-19 in HCWs. Most cases were non-severe (asymptomatic and mild), but over one-third of HCWs were moderate-to-severe cases, a value higher than findings in studies we referred to, mainly due to sample characteristics, such as younger and healthier HCWs in these studies [23,24]. 

Weekly reports from the National Institute of Health in Romania showed that COVID-19 cases among the medical staff overlap only partially on the epidemic curve in the general population, explained by the fact that HCWs represent a segment of the active population (18–65 years) with higher exposure at work and in the community and the differences between adherence to non-pharmacological prevention measures [3]. As the hospital is a reference center for public health alerts, the HCWs are periodically trained regarding self-protective precautions. Before the epidemic’s debut, multiple meetings and workshops were organized periodically to instruct the personnel about protective measures during medical activities. Regarding COVID-19 in different categories of HCWs, our study is concordant with other studies [24,25], showing that the laboratory confirmed group had significantly fewer physicians than the uninfected group, as the activities’ medical profiles differ from other categories regarding exposure. 

The relationship between exposure to COVID-19 patients or the environment was established. Exposure of HCWsin the healthcare setting by direct face-to-face contact activities performed on COVID-19 patients (tracheal intubation or extubation, nebulizer treatment, open airway suctioning, sputum collection, tracheotomy, bronchoscopy, cardiopulmonary resuscitation, and other specified procedures) were not found to be associated with an increased likelihood of the severity of COVID-19. During the high-risk procedures, the HCWs were more careful with protective measures, knowing and understanding the high risk of transmission. Also, the consistent use of personal protective equipment did not tend to be linked to clinical outcomes as the HCWs wore the PPE around confirmed and probable cases. This result supports the importance of using protective equipment for any patient with COVID-19 exposure. A prospective international study established that 1 in 10 HCWs involved in the tracheal intubation of suspected or confirmed COVID-19 cases would have a positive test within the next 21 days, despite adherence to recommended WHO PPE of 87.9% [26]. The data from our study should be interpreted by contextualizing the specifics of the hospital and the non-pharmaceutical measures applied. Excepting the first two months of the study period, the hospital admitted only confirmed COVID-19 cases; all HCWs wore PPE during medical activities and were trained to use the equipment properly. 

The variable exposed and nonexposed based on the activity description was included in the logistic regression (OR 2.652 (1.632–4.31), *p* < 0.001). The results showed that HCWs had an increased risk of moderate-to-severe clinical outcomes if exposed. According to the level of risk, were only a few cases (29, 6.8%) of high risk during medical activities, and this variable was excluded from the logistic regression. However, this finding can be linked to good adherence to personal protective measures. 

As we found that the risk of moderate-to-severe outcomes is higher for HCWs exposed than the HCWs who were not exposed during work tasks, we completed the risk evaluation by searching individual factors (co-morbidities). Variables, such as age, anemia, and obesity, were included in the logistic regression, and the results established that these are good predictors for moderate-to-severe cases, findings concordant with other published studies in all age groups [9,10,12,13,14]. Stratification of individual risk factors in future studies can provide new prediction tools. 

Regarding protection by vaccination, our study established a lower severity in persons vaccinated with two doses, concordant with other studies [27,28]. However, most participants had their first infection with SARS-CoV-2 before the use authorization of COVID-19 vaccines (*n* = 340, 80%). The national vaccination campaign started with the HCWs category in the last week of December 2020. In Romania, there were only four vaccines authorized by the European Medicines Agency (Comirnaty/Pfizer–BioNTech, Spikevax/Moderna Oxford–AstraZeneca, and Janssen/Johnson & Johnson). The HCWs adhered highly to vaccination recommendations, as this group’s national vaccination uptake rate stands at 94% [8]. In INBI, the vaccination of HCWs also started in December 2020; the vaccines used were Comirnaty/Pfizer–BioNTech, Spikevax/Moderna, and Oxford–AstraZeneca; a high adherence to vaccination was also registered. There were no mandatory aspects involved.

Correlations between the clinical outcome and the circulating variant of concern (VOC) did not represent the study’s objective because of the reduced availability of regular sample sequencing. However, we compared nationally reported data on the matter and observed that most of the cases were registered during the circulation of the Wuhan- Hu- 1 variant; additionally, the high severity of clinical outcomes was registered during the circulation of the Wuhan- Hu- 1 variant corresponding to the beginning of the epidemic.

This study has a set of limitations. First, a certain degree of recall bias cannot be excluded because, for the severe cases, the questionnaire was applied after a variable number of days when the patient could answer the phone. However, these potential errors were limited as the information collected in the WHO forms did not influence the results obtained by logistic regression; additionally, the exposure was evaluated based on the current activities of HCWs. All other information included in the statistical analysis (clinical outcome and co-morbidities) was objectively evaluated by the clinician when discharge forms were filled in.

The second limitation is systematic errors by consciously omitting information when applying the questionnaire about the proper use of PPE, as the appropriate use of PPE is mandatory in the hospital. The questionnaire was applied during the epidemiological investigations carried out by the medical staff from the Infection Control Department. They used additional questions to clarify the exposure, and we consider this error minimal. No safety scores were used to assess the subject’s ability to understand the question and to answer it. Still, HCWs were trained on potential exposure within the hospital and specific protection measures. The terminology used in the questionnaire is a known one.

The diagnostic established by the clinician based on clinical or/and imagistic evaluation was the severity criteria used. The length of hospital stay did not represent the severity criterion because of the limitation made by the National methodology for surveillance of COVID-19, which regulated the discharge of COVID-19 patients after a negative test for a certain period at the beginning of the epidemic.

The surveyed population had a high share of women aged between 18 and 65, reflecting the gender distribution and age among the hospital staff. The results should be interpreted in this context.

The strengths of this survey include the high quality of the data collected actively by junior physicians (trainees in field epidemiology) by the team from the hospital’s Department of Infection Control and by medical doctors from the hospital who evaluated the HCWs. The evaluation instrument was a standardized questionnaire recommended by WHO.

## 5. Conclusions

In summary, this study reported milder laboratory-confirmed COVID-19 cases in HCWs than moderate-to-severe cases in a frontline hospital in Bucharest, Romania.

Vaccinated HCWs were protected from severe clinical outcomes, the differences between groups being statistically significant. Risk factors associated with clinical outcomes were exposure to confirmed COVID-19 cases, age, obesity, and anemia.

The insights offered from the study represent a baseline for future studies regarding risk stratification factors and suggest that applying non-pharmacological measures, vaccination to protect HCWs, and knowing their individual risks are essential assets in pandemic preparedness.

## Figures and Tables

**Figure 1 vaccines-11-00899-f001:**
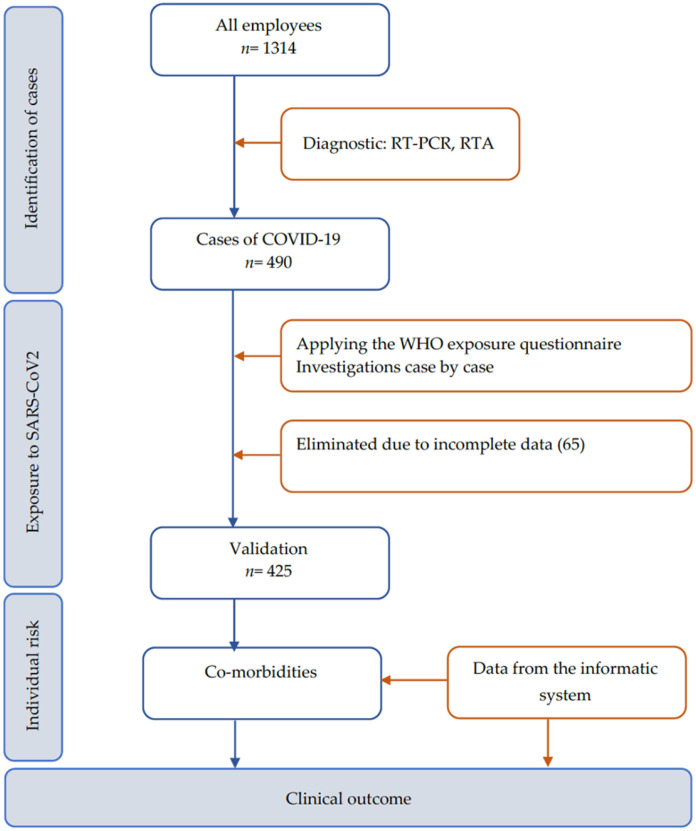
Flow chart of data exclusion for analysis, Bucharest, Romania. Abbreviations: RT-PCR, reverse transcriptase polymerase chain reaction; RTA, rapid test antigen.

**Figure 2 vaccines-11-00899-f002:**
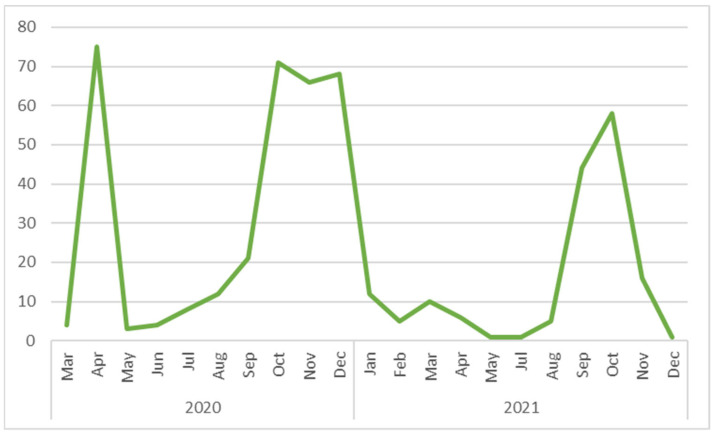
Monthly distribution of COVID-19 cases, in HCWs, in the hospital setting.

**Figure 3 vaccines-11-00899-f003:**
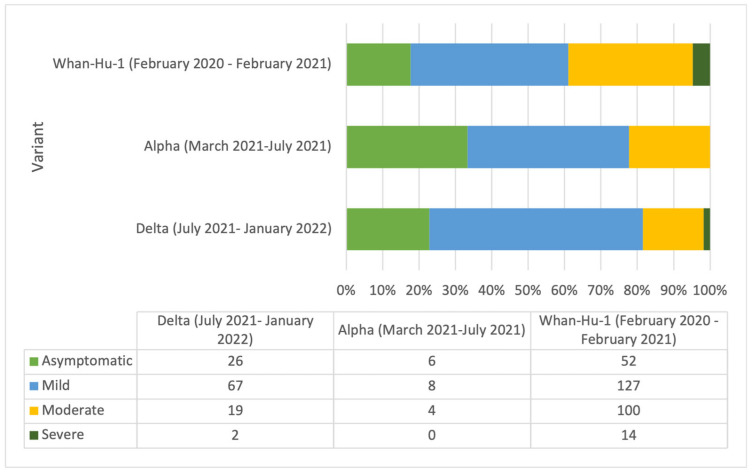
COVID-19 clinical outcome by circulating SARS-CoV-2 variant in Romania.

**Table 1 vaccines-11-00899-t001:** Characteristics of active surveyed HCWs (*n* = 1314) by laboratory evidence of SARS-CoV-2 infection in National Institute for Infectious Diseases “Professor Dr. Matei Balș”, Bucharest, Romania, from 26 February 2020 to 31 of December 2021.

Characteristics		All HCWs *n* = 1314	Laboratory Confirmed Cases *n* = 490 (37.3%)	Uninfected *n* = 824 (62.7%)	*p*-Value
Gender	Female	1104 (84.0%)	417 (85.1%)	687 (83.4%)	*p* = 0.43 ^a^, χ^2^(1) = 0.62
	Male	210 (16.0%)	73 (14.9%)	137 (16.6%)	*p* = 0.42 ^a^, χ^2^(1) = 0.66
Age	Mean (SD)	41.92 (10.5)	43.98 (9.5)	40.70 (10.9)	*p* ≤ 0.0001 ^b^, SE = 0.59
	Median age (years)	42, IQR 33–50	45, IQR 37–51	41, IQR 30–49	*-*
	≤29	232 (17.7%)	51 (10.4%)	181 (22.0%)	*p* ≤ 0.0001 ^a^, χ^2^(1) = 28.39
	30–39	316 (24.0%)	110 (22.4%)	206 (25.0%)	*p* = 0.28 ^a^, χ^2^(1) = 1.14
	40–49	427 (32.5%)	181 (36.9%)	246 (29.9%)	*p* = 0.0088 ^a^, χ^2^(1) = 6.85
	≥50	339 (25.8%)	148 (30.2%)	191 (23.2%)	*p* = 0.0051 ^a^, χ^2^(1) = 7.86
Job category	Nurses	467 (35.5%)	186 (38.0%)	281 (34.1%)	*p* = 0.15 ^a^, χ^2^(1) = 1.99
	Physicians	319 (24.3%)	96 (19.6%)	223 (27.1%)	*p* = 0.0023 ^a^, χ^2^(1) = 9.32
	Healthcare auxiliary activities	359 (27.3%)	148 (30.2%)	211 (25.6%)	*p* = 0.07 ^a^, χ^2^(1) = 3.27
	Other categories	169 (12.9%)	60 (12.2%)	109 (13.2%)	*p* = 0.60 ^a^, χ^2^(1) = 0.26

^a^ Calculated using the χ^2^ test using MedCalc Software Ltd. comparison of means calculator, https://www.medcalc.org/calc/comparison_of_means.php (Version 20.218; accessed 21 February 2023). ^b^ Difference between the means in two independent samples using MedCalc Software Ltd. comparison of means calculator (https://www.medcalc.org/calc/comparison_of_means.php) (Version 20.218; accessed 21 February 2023).

**Table 2 vaccines-11-00899-t002:** Characteristics of COVID-19 cases in HCWs by the clinical outcome in National Institute for Infectious Diseases “Prof. Dr. Matei Balș”, Bucharest, Romania, from 26th February 2020–31 December 2021 (*n* = 425).

Characteristics		All COVID-19 Cases in HCWs *n* = 425(%)	Non-Severe Cases *n* = 279(64.7%)	Moderate-to-Severe Cases *n* = 146(34.3%)	*p*-Value
Gender	Female	362 (85.2%)	237 (84.9%)	125 (85.6%)	*p* = 0.85 ^a^, χ^2^(1) = 0.034
	Male	63 (14.8%)	42 (15.1%)	21 (14.4%)	*p* = 0.85 ^a^, χ^2^(1) = 0.034
Age	Mean (SD)	44.2 (9.41)	43.2 (9.45)	46.0 (9.08)	*p* = 0.0035 ^b^, SE = 0.95
	Median age (years), IQR	45, IQR 37–51	44, IQR 36–50	47, IQR 40–52	
	≤29	42 (9.9%)	33 (11.8%)	9 (6.2%)	*p* = 0.06 ^a^, χ^2^(1) = 3.452
	30–39	92 (21.6%)	67 (24.0%)	25 (17.1%)	*p* = 0.10 ^a^, χ^2^(1) = 2.677
	40–49	161 (37.9%)	101 (36.2%)	60 (41.1%)	*p* = 0.32 ^a^, χ^2^(1) = 0.976
	≥50	130 (30.6%)	78 (28.0%)	52 (35.6%)	*p* = 0.10 ^a^, χ^2^(1) = 2.64
Job category	Nurses	167 (39.3%)	107 (38.3%)	60 (41.1%)	*p* = 0.58 ^a^, χ^2^(1) = 0.303
	Physicians	85 (20.0%)	54 (19.4%)	31 (21.2%)	*p* = 0.64 ^a^, χ^2^(1) = 0.211
	Healthcare auxiliary activities	123 (28.9%)	81 (29.0%)	42 (28.8%)	*p* = 0.96 ^a^, χ^2^(1) = 0.003
	Other categories	50 (11.8%)	37 (13.3%)	13 (8.9%)	*p* = 0.18 ^a^, χ^2^(1) = 1.751
Department	High risk (ICU, ER, COVID-19 wards)	336 (79.1%)	206 (73.8%)	130 (89.0%)	*p* = 0.0003 ^a^, χ^2^(1) = 13.345
	Low risk (other departments)	89 (20.9%)	73 (26.2%)	16 (11.0%)	
Exposure	Yes	268 (63.1%)	159 (57.0%)	109 (74.7%)	*p* = 0.0003 ^a^, χ^2^(1) = 12.816
	No	157 (36.9%)	120 (43.0%)	37 (25.3%)	
Risk categorization after exposure ^c^	High risk	29 (6.8%)	24 (8.6%)	5 (3.4%)	*p* = 0.0445 ^a^, χ^2^(1) = 4.04
	Low risk	396 (93.2%)	255 (91.4%)	141 (96.6%)	
Vaccination	Yes	85 (20.0%)	70 (25.1%)	15 (10.3%)	*p* = 0.0003 ^a^, χ^2^(1) = 13.126
	No	340 (80.0%)	209 (74.9%)	131 (89.7%)	
Co-morbidities	Yes	107 (25.2%)	37 (13.3%)	70 (47.9%)	*p <* 0.0001 ^a^, χ^2^(1) = 61.082
	No	318 (74.8%)	242 (86.7%)	76 (52.1%)	

Abbreviations: ICU, intensive care unit; ER, emergency room. ^a^ Calculated using the χ^2^ test using MedCalc Software Ltd. comparison of means calculator. https://www.medcalc.org/calc/comparison_of_means.php (Version 20.218; accessed 21 February 2023). ^b^ Difference between the means in two independent samples using MedCalc Software Ltd. comparison of means calculator (https://www.medcalc.org/calc/comparison_of_means.php) (Version 20.218; accessed 21 February 2023). ^c^ Risk categorization after exposure according to WHO questionnaire: High risk—the HCW did not respond ‘Always, as recommended’ to questions 5A1–5G, 6A–6F or responded ‘Yes’ to 7A; low risk for COVID-19 virus infection—all other answers.

**Table 3 vaccines-11-00899-t003:** HCWs’ activities performed on COVID-19 patients in the healthcare facility (*n* = 425).

Exposure	All COVID-19 Cases in HCWs *n* = 425 (%)	Non-Severe Cases *n* = 279 (65.7%)	Moderate-to-Severe Cases *n* = 146 (34.3%)	*p*-Value
Direct care to a confirmed COVID-19 patient (yes)	253 (59.5%)	147 (52.7%)	106 (72.6%)	*p* = 0.0001 χ^2^(1) = 15.748
Face-to-face contact (yes)	259 (60.9%)	153 (54.8%)	106 (72.6%)	*p* = 0.0004, χ^2^(1) = 15.748
Aerosol-generating procedures (yes)	74 (17.4%)	45 (16.1%)	29 (19.9%)	*p* = 0.3348, χ^2^(1) = 12.676
Direct contact with the environment of COVID-19 patient (yes)	246 (57.9%)	139 (49.8%)	107 (73.3%)	*p* < 0.0001, χ^2^(1) = 21.6

**Table 4 vaccines-11-00899-t004:** Logistic regression model predicting the likelihood of moderate-to-severe COVID-19 outcomes (*n* = 425).

Factors	Multivariable OR (95%)	*p* Value
Age (years)	1.037 (1.012–1.062)	0.004
Obesity	4.941 (2.462–9.913)	<0.001
Anemia	5.821 (2.402–14.112)	<0.001
Exposure (yes)	2.652 (1.632–4.31)	<0.001

Abbreviation: OR, odds ratio.

## Data Availability

The datasets generated and analyzed during the current study are available from the corresponding author upon reasonable request.

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
