# Peer review of "Vaccination and Factors Related to the Clinical Outcome of COVID-19 in Healthcare Workers—A Romanian Front-Line Hospital’s Experience"

_vaccines, 2023, doi:10.3390/vaccines11050899_

Round 1

Reviewer 1 Report

COVID-19 is still a concern although the epidemic is not as serious as it used to be. The manuscript by Carmen-Daniela Chivu et. al. described the frequency of COVID-19 in healthcare workers (HCWs) in a designated hospital for COVID-19 treatment in Bucharest, Romania and found that there are significant differences between groups for high-risk department, exposure to COVID-19 patients, vaccination, and the presence of co-morbidities. Age, obesity, anemia, and exposure to COVID-19 patients predicted the severity of the clinical outcomes. The manuscript highlight the critical roles of measures to protect HCWs for pandemic preparedness and the response in the event of an outbreak. However, the manuscript is not well written and the data was need check carefully.

Some comments are list below:

1.        As more than 3 years has passed sine the first report of COVID-19, a concise background of the COVID-19 pandemic is preferred in the introduction part. Such as the global infection rate, mortality rate, and current epidemic situation etc., especially the infection situation of medical personnel in other countries.    

2.        There are too many paragraphs in background and conclusion parts, it is more appropriate to summarize it into paragraphs with a theme sentence.

3.        Figure 1, according the description in figure 1 and the main text, the number of validation should be 426 (490-64=426).

4.        The format of the tables should be consistent. The numbers in parentheses in table 1 and table 2 have a “%”, while the numbers in parentheses in table 3 do not.

5.        Table 3, in the first line, “n-425” to “n=425”. 

6.        Line 38, supplement the full name of “ECDC”

7.        Line 51, remove the redundant comma at the end of the sentence.

8.        Line 50 and line 313, “Sars-CoV 2” to “SAR-CoV 2”; “SAR-CoV-2” also appears in the main text, please unify the name.

9.        Line 73, the full name should be provided when RAT first appears.

10.    Line 81, what does “RT-PRCR” mean?  Why is the abbreviation for rapid test antigen RAT instead of RTA?

11.    Line 104, “According” appeared in the middle of the sentence.

12.    Line 111 and line 346, letter “o” appeared in the middle of the sentence.

13.  Line 154-156, “Comparing the frequency of COVID-19 by age group, it was identified that the  disease was less frequent in the age group of 18-29 (10.4% cases vs. 22.0% uninfected, p0.0001a χ2(1) = 28.4) and more frequent in the age group of 40-49 (p= 0.0088a, χ2(1) =  6.85).” In terms of the proportion of positive people, age group of 40-49 is the biggest. This is due age group of 40-49 is also the biggest in all HCWs. From the perspective of positive rate among different age groups, the highest positive rate should be the age group of 50, which is 43.66%(148/339), while the positive rate for the age group of 40-49 is 42.39% (181/427). This difference should be mentioned in the main text, otherwise, the reader may be misled.

14.    Line 194, “p= 0.0037a, SE = 0.95”, while the corresponding data in table 2 is “p= 0.0035b, SE = 0.95”, please confirm.

Author Response

Thank you for your review.

Please find my comments in the attached file.

Reviewer 2 Report

Manuscript (ID: vaccines-2339159) presented results of investigation of factors associated with the clinical outcome of COVID-19 detected through active surveillance in healthcare workers in a tertiary infectious disease hospital in Bucharest, Romania. Several inaccuracies in this manuscript require major revision:

  • Lines 35-39: Cite the appropriate reference for first sentence. Reference numbers 1-4 are missing in this text. Enter the appropriate references that are missing. Also, check and match references in the text and list `References`.

  • Lines 42-45: Provide the appropriate reference for data about COVID-19 in Romania.

  • Line 47: Add a new paragraph in which the results on the frequency of COVID-19 in health care workers across countries should be presented, citing appropriate references.

  • Line 49: Add a new paragraph in which the results on the frequency of risk factors for COVID-19 in health care workers in similar studies should be presented, citing appropriate references.

  • Line 49: Add a new paragraph in which data on the start date of vaccination against COVID-19 in Romania should be presented. Also, provide data on vaccination coverage against COVID-19 in health care workers, as well as vaccination coverage in the general population in Romania.

  • Lines 99-100: Specify whether data collection was conducted by direct interview or was self-reported (by `paper and pencil').

  • Lines 125-130: Suggestion is to transfer this paragraph in section Methods, subsection `2.2. Definition and data collection'.

  • Lines 134-135: Specify what were the criteria for the input of variables in the multivariate logistic regression model.

  • Lines 144-150: Explain the number (Cases of COVID-19; n = 490), and `Applying the study's protocol, 64 cases were excluded from the database due to missing or incomplete data. Rationale: 425 cases are listed in the Validation box. The question is; Is 1 case missing? Explain.

  • Table 2: Check if the first column should have the subtitle ``All COVID-19 cases in HCWs'' and not ``All HCWs''. Correct this. Also, check and correct the number of cases in that subtitle: instead of `n = 1314) write `n = 425`, according to data on Table 1.

  • Lines 211-213: The suggestion is to move this paragraph to Line 197, since that data is presented on Table 2.

  • Lines 214-220: The suggestion is to transfer these results to Line 198, since they are shown on Table 2.

  • Lines 251-265: Suggestion is to move this paragraph to the section Methods, on Line 71.

  • Lines 266-327: Section Discussion should be completely reconstructed. Rationale:
    • Section Discussion has no logical flow;
    • Not all significant results presented in this manuscript have been discussed;
    • There is no comparison of the presented results with the results obtained in similar research, with the citation of appropriate references;
    • Give a possible explanation for the differences in the frequency of COVID-19 in HCWs between the current study and in studies across countries, citing appropriate references;
    • Give a possible explanation for the differences in the representation of risk factors for COVID-19 in HCWs between the current study and in studies across countries, citing appropriate references;
    • Ensure correct citation of all references in this manuscript as a whole, because the references in the text of the paper are cited out of order;
    • Harmonize the number of cited references in the text with the total number of references in the list of References, because the list of References lists a total of 18 references, while only about 10 references are cited in the text of the work. Check and correct all references.

  • Lines 328-352: This text provided quality presentation and discussion of issues that are potential sources of limitations of this study.

  • Lines 358-374: Correct the section Conclusions in order to highlight the most important, statistically significant results in this study (according to multivariate regression analysis).

Author Response

(The authors gave the same response as above.)

Reviewer 3 Report

This is a straightforward and useful questionnaire study. 

Sample: How many total eligbles are there? Is it 1314 or are there other workers?

Only 87.6% of a simple qwuestionnaire contained complete data. In a 5 minute questionnaire this implies imperfect intial supervision of your student data collectors. You could have asked the students to go back and verify the missing data. How do the missing data affect your conclusions?

3.4 Please explain what you mean by "direct logistic regression."

Conclusuions. Your study only explained 14.3% of the variance. Can you explain the remaining variance? Is it partlybecause you bisected your sample?

Minor typos: Figure 1 Data from...is incomplete

Fig 3 Asymptomatic

Author Response

(The authors gave the same response as above.)

Round 2

Reviewer 2 Report

Thank you for the opportunity to re-review manuscript ID: vaccines-2339159. The authors make significant corrections in the revised version of this manuscript and removed several inaccuracies in this manuscript that highlighted in my review, and now it looks like:

  • In revised version, the Introduction section provides sufficient background about COVID-19 in HCWs, as well as data about COVID-19 in HCWs in Romania.
  • Also, this revised manuscript includes the appropriate references for data about COVID-19 in HCWs.  
  • In revised version, the research design is appropriate and the methods adequately described.
  • In revised version, the results are more clearly presented.  
  • The Discussion section is significantly clearer and more complete.
  • The Conclusions are appropriate and supported by the highlighted results.   

I thank the authors for the corrections made in the manuscript, which were making the manuscript more informative and perspicuous for the readers.